# Virulence of *Pseudomonas aeruginosa* in Cystic Fibrosis: Relationships between Normoxia and Anoxia Lifestyle

**DOI:** 10.3390/antibiotics13010001

**Published:** 2023-12-19

**Authors:** Rosanna Papa, Esther Imperlini, Marika Trecca, Irene Paris, Gianluca Vrenna, Marco Artini, Laura Selan

**Affiliations:** 1Department of Public Health and Infectious Diseases, Sapienza University, p. le Aldo Moro 5, 00185 Rome, Italy; rosanna.papa@uniroma1.it (R.P.); marika.trecca@uniroma1.it (M.T.); irene.paris@uniroma1.it (I.P.); laura.selan@uniroma1.it (L.S.); 2Department for Innovation in Biological, Agro-Food and Forest Systems, University of Tuscia, 01100 Viterbo, Italy; imperlini@unitus.it; 3Research Unit of Diagnostical and Management Innovations, Children’s Hospital and Institute Research Bambino Gesù, 00165 Rome, Italy; gianluca.vrenna@opbg.net

**Keywords:** oxygen concentration, antimicrobial resistance, biofilm, pyocyanin, pyoverdine, motility, proteases

## Abstract

The airways of cystic fibrosis (CF) patients are colonized by many pathogens and the most common is *Pseudomonas aeruginosa*, an environmental pathogen that is able to infect immunocompromised patients thanks to its ability to develop resistance to conventional antibiotics. Over 12% of all patients colonized by *P. aeruginosa* harbour multi-drug resistant species. During airway infection in CF, *P. aeruginosa* adopts various mechanisms to survive in a hostile ecological niche characterized by low oxygen concentration, nutrient limitation and high osmotic pressure. To this end, *P. aeruginosa* uses a variety of virulence factors including pigment production, biofilm formation, motility and the secretion of toxins and proteases. This study represents the first report that systematically analyzes the differences in virulence features, in normoxia and anoxia, of clinical *P. aeruginosa* isolated from CF patients, characterized by multi- or pan-drug antibiotic resistance compared to antibiotic sensitive strains. The virulence features, such as biofilm formation, protease secretion and motility, are highly diversified in anaerobiosis, which reflects the condition of chronic CF infection. These findings may contribute to the understanding of the real-world lifestyle of pathogens isolated during disease progression in each particular patient and to assist in the design of therapeutic protocols for personalized medicine.

## 1. Introduction

Cystic fibrosis (CF) is a life-threatening autosomal recessive multiorgan disorder, caused by mutations in the gene encoding the CF transmembrane conductance regulator (CFTR) that mediates chloride transport through the mucus-producing cells [1]. Functional failure of CFTR results in the production of thick and sticky mucus whose retention leads to serious bacterial infections [2]. The most affected organ is the lung whose recurrent chronic infections and local airway inflammation are the main cause of morbidity and mortality in CF patients, who have a life expectancy of 38 years [3,4].

The airways of CF patients are colonized by many pathogens, where the most common is the opportunistic Gram-negative bacterium *Pseudomonas aeruginosa*; in particular, it is the most predominant bacterium in children affected by CF and its prevalence increases with the age of these patients [5]. In general, *P. aeruginosa* is an environmental pathogen able to infect immunocompromised patients, thus accounting for 10% of nosocomial infections [6]. These infections are caused by *P. aeruginosa* developing resistance to conventional antibiotics and represent a major healthcare concern: notably the 12.3% of all patients colonized by *P. aeruginosa* who harbour multi-drug resistant (MDR) species (Cystic Fibrosis Foundation Patient Registry, Annual Data Report 2021). *P. aeruginosa* is highly pathogenic compared to other Gram-negative bacteria, due to the production of several virulence factors [6,7]. These factors, combined with its adaptability, enable *P. aeruginosa* to invade eukaryotic cells and, hence, to colonize CF airways over other pathogens [8]. Therefore, it is not surprising that *P. aeruginosa* survives in the hostile CF lung environment. Although such environments are characterized by viscous and stagnant mucus, providing niches with low levels of oxygen and fluctuating pH, and by the presence of other competitive bacteria and antibiotic molecules, *P. aeruginosa* is able to evade the host immune system [7,8]. To this end, *P. aeruginosa* uses a variety of virulence factors including biofilm formation, motility and the secretion of toxins and proteases [7,9]. These virulence factors allow *P. aeruginosa* persistence in hostile niches of the host where bacterial communities form a mature biofilm surrounded by a robust protective matrix: *P. aeruginosa* growing as biofilm is difficult to eradicate [10]. Many lines of experimental evidence suggest that *P. aeruginosa* growing as biofilm in the lung of CF patients, is subjected to hypoxic or even anaerobic conditions as the chronic infection progresses [11,12,13,14]. This suggests that oxygen concentration may play a key role in the biofilm formation, and in the stability of MDR-*P. aeruginosa* species and their persistence in CF airway mucus.

By contrast, pyocyanin is an organic blue-green pigment produced by *P. aeruginosa* in an oxygen-dependent biosynthetic pathway [15]. It is characterized by a redox-activity-generating reactive oxygen species (ROS), in particular a superoxide, and by inducing cellular oxidative stress. Furthermore, it is able to alter the host immune response with different mechanisms and promote the evasion of the immune response, establishing chronicity of infections [16].

In the present study, we investigated the modulation of *P. aeruginosa* virulence factors in clinical strains isolated from CF patients characterized by multi- (MDR) and pan-drug (PDR) antibiotic resistance profiles compared to antibiotic-sensitive strains (WT), all grown in planktonic and biofilm form under normoxic and anoxic conditions. These findings may contribute to the further understanding of the complex landscape of *P. aeruginosa* phenotype modulation, with particular attention to virulence factors in clinical isolates from CF patients, in relation to their antibiotic resistance profile and to anoxia.

## 2. Results

### 2.1. Pyocyanin Production in P. aeruginosa during CF Infection

Pyocyanin production was analyzed at 24 h and 48 h in normoxia (Appendix A), since its biosynthesis is strongly associated with the presence of oxygen. As expected, its production was strongly strain-dependent but generally it is higher at 24 h rather than 48 h. Moreover, the higher pyocyanin producers were WT (wild type, sensitive strains) and MDR strains (except for PDR4), as reported in Figure 1. The linear decrease in pyocyanin production in the three groups is particularly pronounced at 48 h (Figure 1). The evidenced differences between pyocyanin production for WT and MDR strains at 24 and 48 h were statistically significative (*p* value < 0.01). Differences in PDR strains were not significative. 

### 2.2. Pyoverdine Production in P. aeruginosa during CF Infection

In conditions of poor oxygenation, a lower production of pyoverdine was previously observed [17]. Based on these considerations, the pyoverdine production was assessed only in aerobic conditions at 24, 48 and 72 h. As reported in Appendix A, pyoverdine production was observed mainly after 48 h and 72 h of growth (exceptions are represented by some WT and MDR strains). PDR strains appeared to be weak pyoverdine producers both at 48 h and 72 h. At 72 h there was a notable increase in production by the MDR class, which brings the observed values considerably outside the scale in which the values fall for the remaining part of the bacterial collection. As can be seen in Figure 2, pyoverdine is produced at a very low level from PDR strains, suggesting that it is not a key factor for them. Statistical analyses performed to compare pyoverdine production at different times within the same bacterial class revealed that the data obtained were significative (*p* value < 0.0001 for all conditions, except for MDRs at 24 h compared with 72 h with *p* value < 0.05).

### 2.3. Biofilm Formation in P. aeruginosa during CF Infection

Biofilm formation of bacterial strains derived from CF patients characterized by different antimicrobial profiles was assessed in anoxic and normoxic conditions after 18 h of incubation at 37 °C. All bacterial strains were able to produce biofilm with different capabilities (Appendix A). Biofilm content is independent from the antimicrobial profile of bacterial strains, while significative differences were observed for two tested conditions (anoxia and normoxia). Figure 3 reports the trend of the biofilm amount produced by WT, MDR and PDR. WT-grouped strains were isolated in the early stages of infections and sensitive to antibiotics, while the MDRs and PDRs included isolates in later stages of disease. In normoxic conditions, no large variations were apparent between the three groups, while substantial differences can be observed in anaerobic conditions, where WTs produced a small quantity of biofilm compared with PDRs, and especially compared with MDRs. Differences in biofilm content for each bacterial class in normoxia and anoxia were statistically significative only for WTs (*p* value < 0.05). These data could be indicative for the ability of *P. aeruginosa* clinical strains to adapt to the environment during the progression of infection. In fact, MDR and PDR strains seemed to better adapt to the hostile niches of a fibrotic lung, characterized by reduced oxygen diffusion.

### 2.4. Proteolytic Content in P. aeruginosa during CF Infection

The extracellular proteases secreted by *P. aeruginosa* clinical strains were quantitatively and qualitatively evaluated. Quantitative analysis was performed by azocasein assay. Using azocasein as a substrate it was possible to spectrophotometrically measure the absorbance of the azocompound released by the proteolytic action of the proteases. The concentration of this azocompound is proportional to the concentration of the proteases. As reported in Appendix A, the analysis was performed by analyzing the proteolytic content released into the supernatant after 24 h and 48 h of incubation at 37 °C, in the presence or absence of oxygen. The obtained data from the analysis of bacterial cultures in the presence of oxygen (normoxia) showed a comparable proteolytic activity either at 24 h and 48 h of incubation at 37 °C, for almost all the analyzed clinical strains, with the exception of the strains WT8, PDR6 and PDR8A. In normoxia, we found a content of proteases in the supernatant of WT strains comparable to that of MDR strains, while PDR strains proved to be weak producers of proteases. These data confirm the lower expression of virulence factors in clinical strains with higher antimicrobial resistance. In anoxic conditions, by contrast, the proteolytic content of WT strains showed a notable increase after 24 h of growth. No differences were evidenced for MDR and PDR strains (Figure 4). The differences between proteolytic content in WTs in normoxia and Figure 4 anoxia were statistically significative only at 24 h (*p* value < 0.05). Within the MRD group, the differences evidenced in normoxia and anoxia were statistically significative both at 24 and 48 h (*p* value < 0.0001 at 24 h and *p* value < 0.05 at 48 h, respectively). Conversely, for the PDR group the obtained data were not significative. Overall, these data suggest a higher virulence in anoxic conditions in the first 24 h of infection.

The extracellular proteases secreted by clinical *P. aeruginosa* strains were qualitatively analyzed by using gelatin-zymography. This assay was performed on the culture supernatants of three selected *P. aeruginosa* clinical strains for each bacterial group (high, low and medium protease production for each group): WT2, WT4 and WT5 for WT strains; MDR1, MDR5 and MDR7 for MDR strains; PDR2, PDR5 and PDR7 for PDR strains. Therefore, zymography analysis was performed on the culture supernatants of all these strains at 24 h and 48 h in normoxia or anoxia, and after protein separation on SDS-PAGE gels containing a synthetic substrate (gelatin). After run, SDS-PAGE gels were incubated in a proper activation buffer and stained, thus allowing the detection of the different clear bands corresponding to the proteases/gelatinases active in the bacterial cultures of all the tested clinical strains (Figure 5). Regardless of growth conditions, a gelatin-degrading proteolytic band of approximately 46 kDa was clearly detectable in all clinical strains (Figure 5). However, this gelatinolytic band was considerable in all WT tested strains, in normoxia at 24 h and even more so at 48 h, as well as in anoxia at both time points. The same 46 kDa proteolytic band was also observed in the MDR1, MDR5 and PDR5 strains (Figure 5). Interestingly, the same gelatinolytic band at 46 kDa was just visible in the PRD2 strain in normoxia at 24 h and 48 h, but more visible and active in anoxia (Figure 5). By contrast, 46 kDa gelatinase was more active in MDR7 in normoxia than in anoxia at both time points; whereas in PDR7 it was only active at 48 h both in normoxia and anoxia. At 24 h, this PRD strain, in fact, showed slightly active or inactive protease in normoxia and anoxia, respectively (Figure 5).

On the other hand, the gelatinolytic activities of clinical strains differed for molecular weight > 60 kDa, and it is evident that there is no similarity in their electrophoretic patterns due to the presence or absence of 110 kDa and 77 kDa bands (Figure 5). In conclusion, these data indicate that clinical isolates differentially secrete proteases dependent on their growth phase and condition.

### 2.5. Motility in P. aeruginosa during CF Infection

In the present study we analyzed swimming and swarming motility at 24 h, 48 h and 72 h, in both normoxia and anoxia.

Swimming motility is mediated by a polar flagellum that allows *P. aeruginosa* to move in a liquid medium. In normoxia, all strains except WT8, MDR7, MDR8 and PDR3 showed a swimming motility with different capabilities (Figure 6, left panels). In anoxic conditions, a reduction of swimming motility was apparent for almost all bacterial strains regardless their antimicrobial profiles (Figure 6, right panels).

By optical comparison of the plates grown in the presence or absence of oxygen, phenotypical variations in the colonies were observed. In particular, as expected, in anoxia a reduction of pigments was observed (particularly in strains PA14, WT2, WT3, WT6, WT7, MDR7, MDR8, MDR9, PDR4, PDR5, PDR6).

Swarming motility is defined by the rapid and coordinated translocation of a bacterial population across a semisolid surface. All WT strains, except WT8, showed a swarming motility in the presence of oxygen (Figure 7, left panel). Conversely MDR strains showed a reduced swarming motility except MDR1, MDR2 and MDR9 strains; in PRD strains only PDR5, PDR6 and partially PDR8A showed swarming activity (Figure 7, left panel). This kind of motility was completely inhibited in anoxic conditions, where only WT3 showed a very slight swarming motility. Diameters of swimming and swarming halos were reported in Appendix A section.

## 3. Discussion

During chronic infection of CF airways, *P. aeruginosa* adapts its phenotype within the specific ecological niches of the altered lung parenchyma. Thanks to its high plasticity, it displays several mechanisms that enable it to persist in an ecological niche characterized by low oxygen concentration, nutrient limitation, high osmotic pressure and oxidative stress, and in competition with other microorganisms.

These modifications can occur early or late during the progression of the colonization [18]. For example, flagella and pili allow bacterial cells to reach and establish themselves in the pulmonary niche in the early phase of infection, but their expression is afterwards downregulated in order to activate a more complex machinery able to guarantee its survival in the inhospitable environment of CF lung [19].

Changes in other virulence factors such as an increase in antimicrobial resistance and/or modifications in the biofilm lifestyle, protease secretion, pyocyanin and pyoverdine production, are also involved in the transition to chronic pathophenotype. This mechanism of adaptation is also reflected on surface protein pattern, where substantial differences appeared in chronic antibiotic resistance of *P. aeruginosa* isolates compared to early antibiotic sensitive *P. aeruginosa* strains [20].

CF progression leads also to a considerably reduced oxygen concentration in the lungs; as a consequence, some bacteria switch to anaerobic metabolism while infection shifts towards chronicity, especially during late-stage disease [12]. Furthermore, a hypoxic microenvironment increases bacterial multidrug resistance by elevating the expression of multidrug efflux pumps, as recently reported [21]. In particular, antibiotic efflux pumps are modulated by two-component systems, such as the RstA/RstB system. Regulator RstA positively regulates the enzymes involved in the anaerobic nitrate respiratory chain. On the other hand, overexpression of efflux pumps leads to enhanced consumption of oxygen, causing microenvironmental hypoxia, which in turn promotes anaerobic nitrate respiration in *P. aeruginosa* [21].

A previous study evaluated how the transition from a condition of normoxia to anoxia can influence the coexistence of *S. aureus* and *P. aeruginosa* in the lungs of CF patients, and concluded that this switch drives the two pathogens to colonize different regions of lung [22].

In this study, we systematically analyzed the differences in virulence features of clinical *P. aeruginosa* strains characterized by multi- and pan-drug antibiotic resistance profiles compared to antibiotic sensitive strains. These virulence features, such as biofilm, protease secretion and motility, are highly diversified in anaerobiosis, which reflects the condition of CF chronic infection.

Firstly, we characterized pigment production along the clinical bacterial strains. Pyocyanin is a secondary metabolite secreted by *P. aeruginosa* in aerobic conditions, which can undergo a redox cycle resulting in the generation of reactive oxygen species (ROS) harmful to the host cell [23]. For this reason, it was only assessed in the presence of oxygen. As expected, pyocyanin production decreased in PDR and MDR strains compared to WT ones at 24 h and mainly at 48 h of growth (Figure 2). This latter finding is strongly supported by previous reports in the literature, where pyocyanin synthesis appears to be fundamental in the early stages of infection until the establishment of chronic infections [24]. Once bacteria have adapted to the host, in fact, they reduce the expression of virulence factors to escape the attention of the immune system.

No significative differences were observed in pyoverdine production among WT, MDR and PDR groups, except for four MDR strains whose production at 72 h of growth was at least one order of magnitude higher than that observed in the other analyzed strains (Figure 3). Pyoverdine is a fluorescent siderophore produced by *Pseudomonas* species, with the function of intracellular iron acquisition [25]. Furthermore, it induces an iron accumulation that is toxic for eukaryotic cells. In fact, the elevated reactivity of the soluble Fe^2+^ ion for hydrogen peroxide and for oxygen, in general, leads to the production of highly reactive oxygen species (ROS), which culminates in the damage of essential cellular components causing oxidative stress and cellular death [26]. Pyoverdine is usually implicated in acute illness, but also in the production of mature biofilms, so its role in the progression of the disease is still unclear [27].

Regarding biofilm formation, WT, MDR and PDR strains did not show significative differences when they were grown in the presence of oxygen; conversely in anoxia, while WT strains reduced their ability to form biofilm, MDR and PDR strains showed a significative increase in the amount of biofilm (Figure 3). This observation could be explained with the adaptation of multi-resistant strains to persist and colonize the CF lung in the absence of oxygen, due to the thick mucus, which occurs after the onset of bacterial infection. Conversely, this mechanism of adaptation seems to be adopted less in WT strains isolated from early infections, possibly because bacterial cells still colonize microenvironments where anoxia is not extended.

Protease expression and activity are known to depend on bacterial growth phases and/or conditions, thus contributing to specific environmental adaptations of bacteria that lead to differences in antimicrobial profiles [28,29]. As expected, the secretion of proteases was higher in the WT group derived from early infection. In this condition, bacteria still express virulence factors to counteract the host defenses. This is mainly evidenced in anoxia, a typical condition of a chronic infection where bacteria are usually multi-resistant rather than antibiotic-sensitive strains. However, protease quantitative data, although unexpected, were in line with those obtained from gelatin-zymography. The main proteolytic band at 46 KDa, visible in all tested clinical strains, was more active in the WT group in normoxia, as well as in anoxia. In addition to the similarities in the proteolytic patterns among the three groups of bacteria, the qualitative analysis of clinical isolates also revealed differences within the same group or depending on the growth phase and condition (Figure 5).

With regard to bacterial motility, however, it is well known that bacteria lose the ability to express the flagellum when infection progresses from acute to chronic form. This trend was mainly highlighted when bacteria were grown in anoxia (especially for the MDR and PDR strains). However, there are exceptions, represented for example, by the PDR5 and PDR8A strains, that are still able to move. Swarming motility, on the other hand, was completely absent in anoxia for almost all bacterial strains.

In conclusion, our data suggest that it is fundamental to analyze the virulence factors produced by clinical *P. aeruginosa* during CF progression rather than in conventional laboratory conditions characterized by an optimal oxygen tension. By contrast, we think it is mandatory to analyze them in a condition as close as possible to the real condition of a CF lung: a highly hostile environment characterized by largely anoxic regions, where bacteria adapt their virulence arsenal to persist and escape the immune system response.

## 4. Materials and Methods

### 4.1. Ethics Approval and Informed Consent

This study was approved by the Ethical Committee of Pediatric Hospital and Institute of Research Bambino Gesù (OPBG) in Rome, Italy (No. 1437_OPBG_2017 of July 2017). The study was conducted in respect of the Declaration of Helsinki as statement of ethical principles for medical research involving human subjects. All participants, or the legal guardians of those included in the study, signed an informed consent form.

### 4.2. Bacterial Strains and Growth Conditions

Clinical strains of *P. aeruginosa* were isolated from airways of CF patients in follow-up to OPBG. Bacterial strains were classified as follows: (i) nine antibiotic-sensitive strains conventionally classified as wild type (WT) were isolated from CF patients with recent infections (<12 months); (ii) nine multi-drug resistant non-mucoid strains (MDR) were isolated from CF patients with chronic colonization (4–15 years); (iii) seven pan-drug resistant non-mucoid strains (PDR) were isolated from CF patients with chronic colonization (4–15 years). These strains were previously classified for their antimicrobial profiles in Montemari et al. [20]. Reference strain *P. aeruginosa* PA14 was used. According to The European Committee on Antimicrobial Susceptibility Testing (http://www.eucast.org, accessed on 20 October 2023), WT strains are defined as sensitive to all antimicrobials, while MDR strains are resistant to at least one agent in three or more antimicrobial categories and PDR strains are resistant to all antibiotics in all classes. Antimicrobial profiles for all tested strains are summarized in Table 1. Bacteria were grown in Brain Heart Infusion broth (BHI, Oxoid, Basingstoke, UK). Planktonic condition was performed at 37 °C under orbital shaking (180 rpm), while biofilm formation was performed at 37 °C in static conditions. For anaerobiosis Ruskinn Concept 400 Workstation was used (LabTech, Heathfield, UK).

### 4.3. Biofilm Formation

The biofilm quantification was assessed by microtiter plate (MTP) biofilm assay [30]. An overnight bacterial culture was 1:100 diluted into BHI fresh medium and aliquoted in the wells of a sterile 96-well polystyrene flat base plate. The plates were overnight incubated at 37 °C under static conditions in aerobic and anaerobic conditions. After incubation, the supernatant containing planktonic cells were gently removed and the plates were washed with double-distilled water. Then the microtiter plates were patted dry in an inverted position. The staining was performed with 0.1% crystal violet for 15 min at room temperature. The excess of crystal violet was removed by washing the wells with double-distilled water. The microtiter plates were thoroughly dried. The remaining biofilm was dissolved with 20% (*v*/*v*) glacial acetic acid and 80% (*v*/*v*) ethanol, and spectrophotometrically measured at 590 nm. Each experiment was performed in 6-replicates, and each data point was composed of four independent experiments.

### 4.4. Pyocyanin Assay

Pyocyanin production was determined as previously described [31]. Briefly, bacterial cells were inoculated in BHI broth and aerobically incubated for different times at 37 °C. The cell-free supernatant, recovered after centrifugation at 10,000 rpm for 15 min, was used for pyocyanin extraction. The supernatant was mixed in a 1:1 ratio with chloroform. The mixture was inverted and then decanted at room temperature up to the separation between the two phases. Pyocyanin (lower phase) was transferred into a new tube and the same volume of 0.2 M HCl was added. The mixture was inverted and decanted to allow the separation of the two phases. The upper pink layer, containing pyocyanin, was recovered and quantified at 520 nm. Pyocyanin content was normalized for the optical density of the corresponding bacterial culture.

### 4.5. Pyoverdine Assay

For pyoverdine quantification, bacteria were grown in King’s B Medium supplemented with 0.5% (*w*/*v*) of Casamino Acid (CAA) at 37 °C in aerobic and anaerobic conditions [31]. Pyoverdine was quantified by reading 100 μL of each *P. aeruginosa* cell-free supernatant into a black 96-well plate (Greiner, Stonehouse, UK) at excitation and emission wavelengths of 400/460 nm, as previously reported [32], on an Infinite 200 PRO (Molecular Devices, San Jose, CA, USA) fluorescence microplate reader. The background level of fluorescence was measured using the same medium. Each measurement was normalized for the optical density detected in each bacterial culture.

### 4.6. Protease Assay

The extracellular proteolytic activity of *P. aeruginosa* was determined by azocasein assay [33]. Cell-free culture supernatant (150 μL) was added to a solution containing 500 μL of 0.3% *w*/*v* azocasein (Sigma, St. Louis, MO, USA) in 50 mM Tris–HCl, 0.5 mM CaCl_2_ pH 7.5 and incubated at 37 °C for 30 min. To stop the reaction, 650 μL of l0% ice-cold trichloroacetic acid was added. The obtained mixture was incubated at 4 °C for 10 min. Then, the insoluble azocasein was removed by centrifugation at 10,000 rpm for 10 min and the supernatant was measured at OD 400 nm.

### 4.7. Zymography Assay

Assay was performed on culture supernatants of *P. aeruginosa* clinical strains. Unconcentrated culture supernatants (20 μL) were combined with Laemmli sample buffer without reducing agent or boiling the protein samples [33]; these were then separated on a 10% SDS-PAGE gel containing 0.2% gelatin (Sigma–Aldrich, Milan, Italy) with a 4% polyacrylamide in stacking gel. Molecular masses of protease bands were estimated by using pre-stained molecular mass markers (mPAGE Color Protein Standard, Millipore, Milan, Italy). After electrophoretic run, gels were incubated, at room temperature, with 2.5% Triton X-100 for 1 h, thus removing SDS and renaturing the proteins. After two washes in distilled water, gels were incubated overnight at 37 °C in a development buffer, as previously described [33]. Gels were stained for 45 min with 0.5% Coomassie blue R-250 in glacial acetic/methanol/distilled water (1:3:6), and destained in distilled water. The zymogram images, containing clear bands on a blue background, were acquired by using ChemiDoc XRS+ System (Biorad, Segrate, Italy). Zymogram experiments were repeated at least twice.

### 4.8. Motility Assays

#### 4.8.1. Swarming Assay

The swarming assay was performed as previously published by Yang and coworkers [34], with some modifications [31]. Plates after the inoculation, were incubated at 37 °C for 24 h, 48 h and 72 h in normoxia and anoxia. After the incubation period, plates were photographed, and halos were measured. Swarming assays were repeated three times.

#### 4.8.2. Swimming Assay

The swimming assay was conducted in accordance with previous research [34], with some modifications [31]. Plates after the inoculation, were incubated at 37 °C for 24 h, 48 h and 72 h in normoxia and anoxia. After the incubation period, plates were photographed, and halos were measured. Swarming assays were repeated three times.

## 5. Conclusions

The novelty of this work is the systematic analysis of the virulence factors displayed by clinical strains in growth conditions that reflect the complexity of the hostile lung environment in which they grow and cause tissue damage. Our data show the variability of virulence factors expressed by *P. aeruginosa* during CF progression. They seem to be unpredictable so far, probably because of the number of interacting variables that modulate bacterial phenotype in such extreme microenvironments. Further studies would be useful to understand the real-world lifestyle of the pathogens isolated during disease progression in each individual patient. The identification of the most significant bacterial biomarkers, and their analysis/integration (possibly also with biomarkers of the patient), as the input for Artificial Intelligence algorithms could allow the design of therapeutic protocols of personalized medicine. This kind of approach is increasingly demanded by clinicians, mainly for chronic and multifactorial diseases, such as tumors and chronic infections, where therapy failure is a frustrating experience.

## Figures and Tables

**Figure 1 antibiotics-13-00001-f001:**
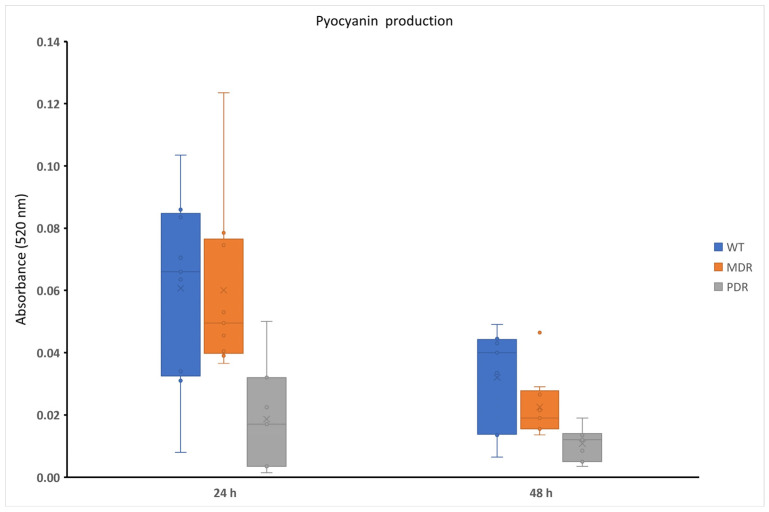
Pyocyanin production of WT, MDR and PDR strains at 24 h and 48 h in normoxia. WT: wild type, sensitive strains; MDR: multi-drug resistant strains; PDR: pan-drug resistant strains.

**Figure 2 antibiotics-13-00001-f002:**
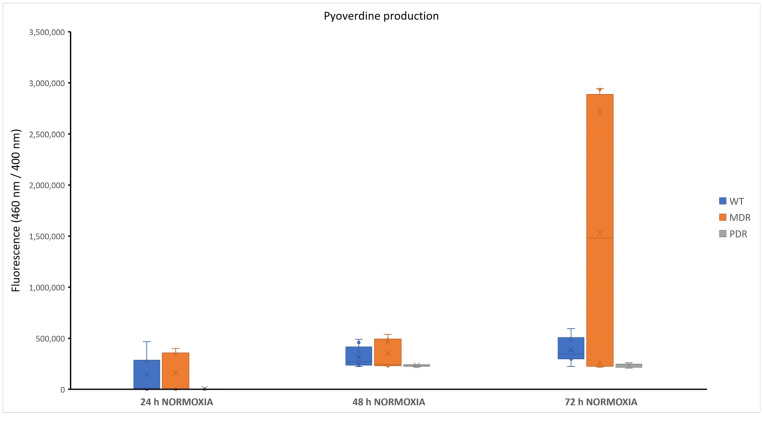
Pyoverdine production of WT, MDR and PDR strains at 24 h, 48 h and 72 h. WT: wild type, sensitive strains; MDR: multi-drug resistant strains; PDR: pan-drug resistant strains.

**Figure 3 antibiotics-13-00001-f003:**
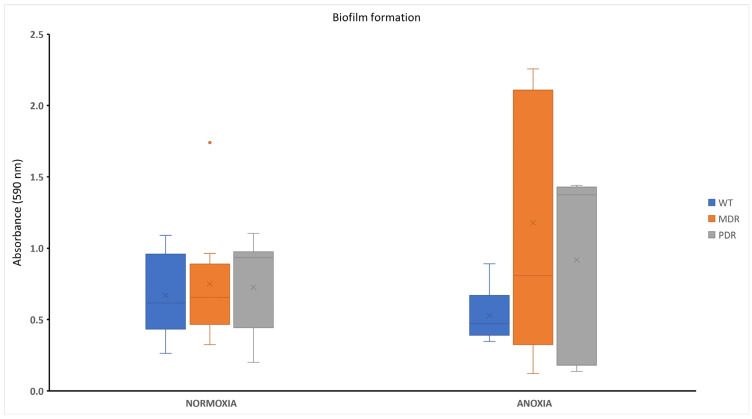
Biofilm formation of WT, MDR and PDR strains in normoxia and anoxia. WT: wild type, sensitive strains; MDR: multi-drug resistant strains; PDR: pan-drug resistant strains.

**Figure 4 antibiotics-13-00001-f004:**
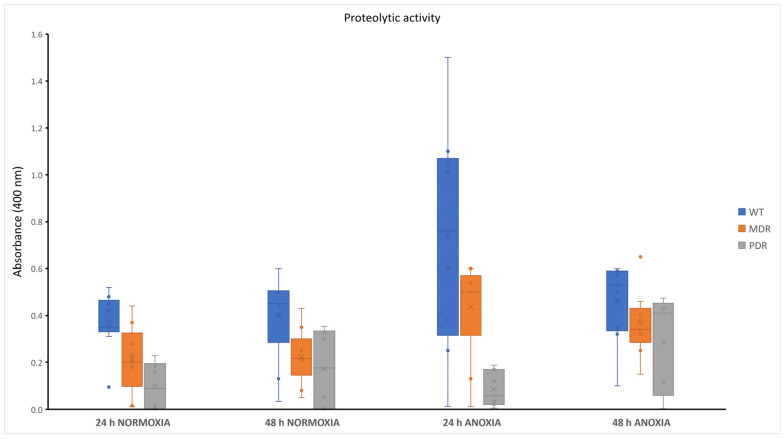
Protease production of WT, MDR and PDR strains at 24 h and 48 h in normoxia and anoxia. WT: wild type, sensitive strains; MDR: multi-drug resistant strains; PDR: pan-drug resistant strains.

**Figure 5 antibiotics-13-00001-f005:**
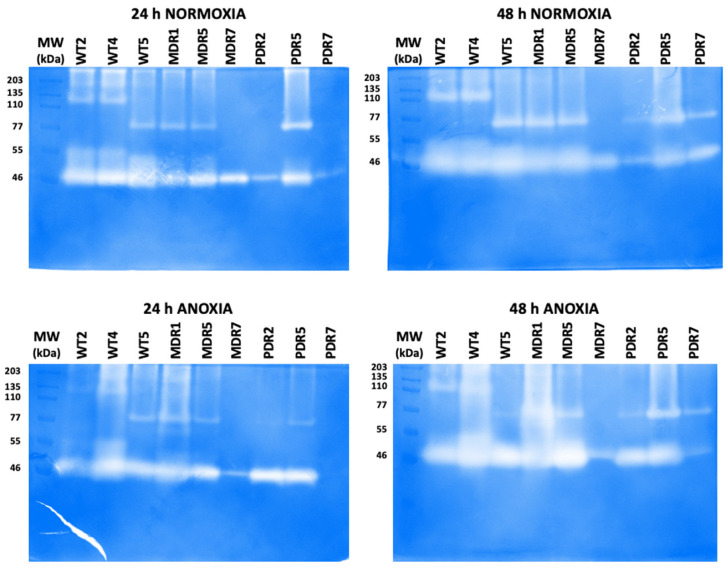
Gelatin-zymography to assay proteases secreted by *P. aeruginosa* clinical strains at 24 h and 48 h in normoxia and anoxia. Protein samples from unconcentrated culture supernatants were separated by using 10% SDS-PAGE gel with 0.2% gelatin. All gels were incubated in a development buffer and destained with 0.5% Coomassie blue R-250, thus obtaining clear bands corresponding to active proteases on a blue background. MW, molecular weight in kDa of protein markers. WT: wild type, sensitive strains; MDR: multi-drug resistant strains; PDR: pan-drug resistant strains.

**Figure 6 antibiotics-13-00001-f006:**
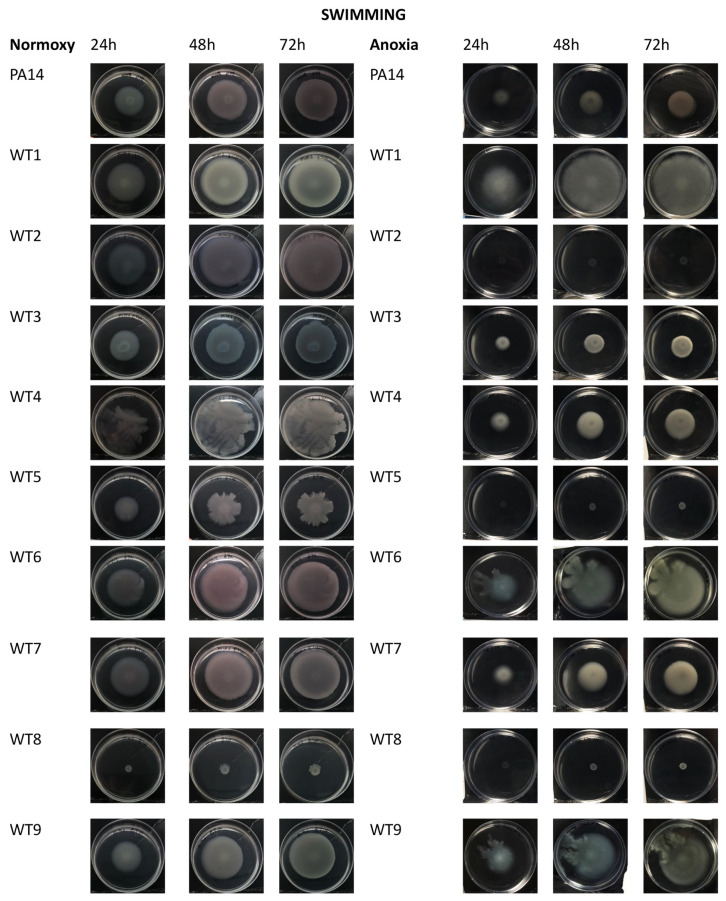
Motility assay: swimming assay of *P. aeruginosa* bacterial strains in normoxia (left panel) and anoxia (right panel) measured at 24 h, 48 h and 72 h of bacterial growth. WT: wild type, sensitive strains; MDR: multi-drug resistant strains; PDR: pan-drug resistant strains.

**Figure 7 antibiotics-13-00001-f007:**
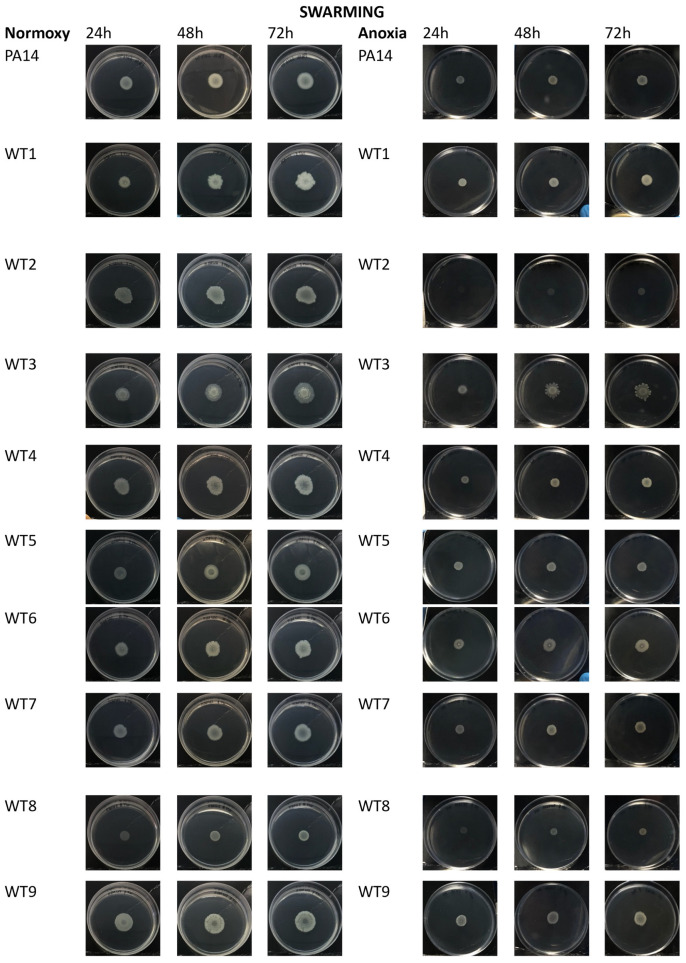
Swarming motility of *P. aeruginosa* bacterial strains in normoxia (left panel) and anoxia (right panel) measured at 24 h, 48 h and 72 h of bacterial growth. WT: wild type, sensitive strains; MDR: multi-drug resistant strains; PDR: pan-drug resistant strains.

**Table 1 antibiotics-13-00001-t001:** Antimicrobial profiles of clinical and PA14 strains [20].

Strain	Fluoroquinilones		Penicillins	Monobactams	Cephalosporins				Aminoglycosides		Carbapenems	
	CIP	LEV	TZP	ATM	CAZ	FEP	CZA	C/T	AK	TOB	IM	MRP
	5 µg	5 µg	30–6 µg	30 µg	10 µg	30 µg	10–4 µg	30–10 µg	30 µg	10 µg	10 µg	10 µg
PA14	S	S	S	S	S	S	-	-	S	S	S	S
WT1	I	R	I	I	I	I	-	-	S	S	I	S
WT2	I	I	I	I	I	I	-	-	S	S	I	S
WT3	I	I	I	-	I	I	S	S	S	S	I	S
WT4	I	I	I	-	I	I	S	S	S	S	I	S
WT5	I	I	S	-	I	I	-	-	S	I	-	S
WT6	I	I	I	-	I	I	S	S	S	S	I	S
WT7	I	I	I	I	I	I	-	-	S	S	I	S
WT8	I	I	I	-	I	I	S	S	S	S	I	S
WT9	I	I	I	-	I	I	S	S	S	S	I	S
MDR1	R	R	R	R	R	R	R	R	R	R	R	R
MDR2	I	R	R	R	R	R	R	R	R	R	R	R
MDR3	I	R	R	R	R	R	S	R	R	R	R	I
MDR4	I	R	R	R	R	R	S	S	R	R	R	I
MDR5	R	R	R	R	R	R	S	S	R	R	R	R
MDR6	R	R	S	S	S	S	-	-	R	R	R	R
MDR7	R	R	R	R	R	R	S	S	R	R	R	R
MD8	R	R	R	R	R	R	S	R	S	S	R	I
MDR9	R	R	I	R	S	R	S	S	R	R	I	R
PDR2	R	R	R	I	R	R	R	R	R	R	R	R
PDR3	R	R	R	R	R	R	-	-	R	R	R	R
PDR4	R	R	R	R	R	R	R	R	R	R	R	R
PDR5	R	R	R	R	R	R	R	R	R	R	R	R
PDR6	R	R	R	R	R	R	R	R	R	R	R	R
PDR7	R	R	R	R	R	R	R	R	R	R	R	R
PDR8A	R	R	R	R	R	R	R	R	R	R	R	R

Antimicrobial susceptibility was performed according to the guidelines of EUCAST Clinical Breakpoint Tables v. 13.0 (valid from 1 January 2023). CIP: ciprofloxacin; LEV: levofloxacin; TZP: Piperacillin-tazobactam; ATM: aztreonam; CAZ: ceftazidime; FEP: cefepime; CZA: Ceftazidime/avibactam; C/T: ceftolozane/tazobactam; AK: amikacin; TOB: tobramycin; IM: imipenem; MRP: meropenem; S: sensitive, R: resistance; I: intermediate, -: not tested.

## Data Availability

Data may be obtained from the appropriate author depending on which data is requested.

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
