# Peer review of "Virulence of Pseudomonas aeruginosa in Cystic Fibrosis: Relationships between Normoxia and Anoxia Lifestyle"

_antibiotics, 2023, doi:10.3390/antibiotics13010001_

Round 1

Reviewer 1 Report

Comments and Suggestions for Authors

1-    Authors should modify the keywords to not be a repetition of the title.

2-    The first paragraph in the results (lines: 75-80) should be transferred to the introduction.

3-    The first paragraph in page 4 (Lines: 6-11) should be added to discussion.

4-    In page 11 figure 11 did not mentioned also in figures (figure 11 is missing or miss named 12).

5-    The title of figure 10 and 11 is the same and should be revised, I think the first is for motility and the second for swarming.

6-    Conclusion is missing and can be added through deducing the last paragraphs in discussion.

Author Response

Attached is the response to reviewer 1

Reviewer 2 Report

Comments and Suggestions for Authors

In my assessment, this manuscript contains an excess of figures in the main text. I propose relocating Figures 1, 3, 5, and 7 to the Supplementary section. Additionally, the authors should ensure that readers can clearly discern differences in the representation of identical phenotypic data. This clarity is currently lacking, as seen in the legends of Figures 3 and 4.

Statistical data, as the authors have provided for the biofilm data, should be performed and incorporated in the graphs to improve interpretation, at least for the 24h vs 48h comparisons.

Figures 10 and 11 would be better suited for the Supplementary Data section. A new graphical representation comparing halo measurements for the tested conditions should be created. This adjustment allows readers to assess the size of halo differences objectively, eliminating reliance on visual observation alone. Concerning the swimming and swarming assay, the authors should explicitly mention the number of tested replicates and account that into the statistical analysis of the halos.

In addition to these major points, I have a few minor comments:

- In line 176, there seems to be a typographical error; it should be "displays."

- Line 362 contains a duplicated sentence.

Author Response

Attached is the response to reviewer 2

Reviewer 3 Report

Comments and Suggestions for Authors

- This study  investigated the modulation of virulence factors in P. aeruginosa clinical strains isolated from CF patients and characterized by multi- and pan-drug  antibiotic resistance profiles compared to antibiotic-sensitive strains, all grown in plank-68 tonic and biofilm form under normoxic and anoxic conditions.

-However, this study is well designed. Materials and methods are formal and well  designed. The manuscript should be subjected to English editing to improve the language and easy to read. 

- Comments:

Line 24: Please delete in

Line 30: deriving should be derived

Line 33: is independent of : replace of with from

Line 41: indicative of should be indicative for

Line 127: delete to

Line 131: delete of

Line 167: displays not diplay

Line 186: delete both in (in protease, in pyocyanin)

Line 277: delete (2022) should be Montemari et al. [25]

Line 278: delete as

Line328: it is better not start with number you can write Cell culture supernatant (150ul)

Comments on the Quality of English Language

The manuscript should be subjected to English editing to improve the language and easy to read. 

Author Response

Attached is the response to reviewer 3

Reviewer 4 Report

Comments and Suggestions for Authors

Virulence of Pseudomonas aeruginosa in cystic fibrosis: relationships between normoxia and anoxia lifestyle by Papa et al.

The paper reports the influence of normoxia and anoxia state on expression of some properties that could affect virulence of Pseudomonas isolates differing in  antibiotic resistance. The highest value of these results is they can be an indication for medical treatment of CF and to design a guideline for  personalized  medicine.

Is possible to indicate the effects of both states (normoxia and anoxia) on the expression of antibiotic resistance especially since the relevant publication is cited: “Furthermore, hypoxic microenvironmental increases bacterial multidrug resistance by elevating the expression of multidrug efflux pumps, as recently reported [26].”

Figures 10 and 11 should be placed in supplementary

L 75-80 These are not results, move this fragment of text to the introduction.

L 84 What does the abbreviation WT mean - wild type?

L 85-87 This latter is strongly supported by previous reports in literature, where pyocyanin synthesis appears to be fundamental in the early stages of infection until the establishment of chronic infections [17].

These are not the authors' own results - move this sentence to the introduction or discussion

Figures 2, 3, 4, 6 and 8 ….explain the abbreviations WT, MDR and PDR

From page 4 there is no continuation of text line numbering

L 5-12

2.2. Pyoverdine production in P. aeruginosa during CF infection

Pyoverdine is a fluorescent siderophore produced by Pseudomonas species, functional to the intracellular iron acquisition [18]. Furthermore, it induces an iron accumulation which is toxic for eukaryotic cells. In fact, the elevated reactivity of the soluble Fe 2+ for hydrogen peroxide and, in general, for oxygen leads to the production of highly reactive oxygen species (ROS), which culminates in the damage of essential cellular components causing oxidative stress and cellular death [19]. However, in conditions of poor oxygenation, a poor production of this siderophore was observed [20].

These are not results, move this fragment of text to the introduction.

L 19 As observable in Fig. 4, pyoverdine is not a key virulence factor for PDR strains.

The figure shows that pyoverdine is not produced or is produced at a low level

L 66-67 These data suggest a higher virulence in hypoxic conditions in the first 24 h of infection.

rather, higher expression of virulence-related factors

but the hypoxic state is not the same as anoxia, in Figure 6 there is the inscription anoxia

L 92-94 Protease expression and activity are known to depend on bacterial growth phases and/or conditions, thus contributing to specific environmental adaptations of bacteria that lead to differences in antimicrobial profiles [21, 22].

These are not the authors' own results - move this sentence to the introduction or discussion

L 138 and L 140 (Fig. 12, left panel) or Fig. 11

Figures 10 and 11 are signed the same way.

L 160 Figure 10. Motility assay: swimming assay of P. aeruginosa bacterial strains in normoxia (left panel) and anoxia (right panel) measured at 24 h, 48 h and 72 h of bacterial growth.

L 171 Figure 11. Motility assay: swimming assay of P. aeruginosa bacterial strains in normoxia (left panel) and anoxia (right panel) measured at 24 h, 48 h and 72 h of bacterial growth.

Figures 10 and 11 are too large and in my opinion they can be moved to supplementary data

L 293 in biofilm assay what was the reference

Why BHI medium for Pseudomonas cultivation was used

L 336 Zimography

L 361 362 are repeated

Author Response

Attached is the response to reviewer 4

Round 2

Reviewer 3 Report

Comments and Suggestions for Authors

Thanks for authors for adequte reply to revision. All comments were done.

I suggest publication of the manuscript